# The Biomedical Importance of the Missing Pathway for Farnesol and Geranylgeraniol Salvage

**DOI:** 10.3390/molecules27248691

**Published:** 2022-12-08

**Authors:** Ignasi Bofill Verdaguer, Marcell Crispim, Agustín Hernández, Alejandro Miguel Katzin

**Affiliations:** 1Department of Parasitology, Institute of Biomedical Sciences of the University of São Paulo, Av. Lineu Prestes 1374, São Paulo 05508-000, Brazil; 2Integrated Unit for Research in Biodiversity (BIOTROP-CCBS), Center for Biological and Health Sciences, Federal University of São Carlos, São Carlos 13565-905, Brazil

**Keywords:** farnesol, isoprenoids, diseases, cancer, dyslipidemias, geranylgeraniol, salvage pathway, missing pathway, fosmidomycin, bisphosphonates, statins, polyprenol kinase, polyprenyl phosphate kinase

## Abstract

Isoprenoids are the output of the polymerization of five-carbon, branched isoprenic chains derived from isopentenyl pyrophosphate (IPP) and its isomer, dimethylallyl pyrophosphate (DMAPP). Isoprene units are consecutively condensed to form longer structures such as farnesyl and geranylgeranyl pyrophosphate (FPP and GGPP, respectively), necessary for the biosynthesis of several metabolites. Polyprenyl transferases and synthases use polyprenyl pyrophosphates as their natural substrates; however, it is known that free polyprenols, such as farnesol (FOH), and geranylgeraniol (GGOH) can be incorporated into prenylated proteins, ubiquinone, cholesterol, and dolichols. Furthermore, FOH and GGOH have been shown to block the effects of isoprenoid biosynthesis inhibitors such as fosmidomycin, bisphosphonates, or statins in several organisms. This phenomenon is the consequence of a short pathway, which was observed for the first time more than 25 years ago: the polyprenol salvage pathway, which works via the phosphorylation of FOH and GGOH. Biochemical studies in bacteria, animals, and plants suggest that this pathway can be carried out by two enzymes: a polyprenol kinase and a polyprenyl-phosphate kinase. However, to date, only a few genes have been unequivocally identified to encode these enzymes in photosynthetic organisms. Nevertheless, pieces of evidence for the importance of this pathway abound in studies related to infectious diseases, cancer, dyslipidemias, and nutrition, and to the mitigation of the secondary effects of several drugs. Furthermore, nowadays it is known that both FOH and GGOH can be incorporated via dietary sources that produce various biological effects. This review presents, in a simplified but comprehensive manner, the most important data on the FOH and GGOH salvage pathway, stressing its biomedical importance The main objective of this review is to bring to light the need to discover and characterize the kinases associated with the isoprenoid salvage pathway in animals and pathogens.

## 1. Isoprenoid Biosynthesis and Distribution

Isoprenoids are the most widespread and diverse group of compounds in nature. They are produced by all organisms and are also thought to be some of the most ancient lipids, originating as constituents of the primitive membranes in the first living organisms [1]. All isoprenoids are made up of five carbons; branched isoprenic chains, derived from isopentenyl pyrophosphate (IPP); and its isomer, dimethylallyl pyrophosphate (DMAPP). Isoprenoids may be synthesized by the mevalonate pathway (MVA pathway) or the non-MVA pathway, also known as the methyl erythritol phosphate pathway (MEP pathway). These are the only well-characterized pathways for IPP/DMAPP production [2,3]. However, there are pieces of evidence for some leucine-dependent pathways which are still not well-characterized [4,5]. These lie outside the scope of this review and, therefore, will be dealt with no further.

The MVA pathway was the first isoprenoid biosynthesis pathway discovered. It is active in plant cytosol, fungi, archaebacteria, eubacteria, and some protozoa groups [2,6]. This pathway starts with the condensation of an acetyl-coenzyme A (acetyl-CoA) molecule with acetoacetyl-CoA to yield 3-hydroxy-3-methyl-glutaryl-CoA (HMG-CoA), a reaction catalyzed by HMG-CoA synthase. Then, the enzyme HMG-CoA reductase (HMGR), produces MVA. This enzymatic step can be inhibited by statins (e.g., simvastatin, lovastatin, atorvastatin, or pitavastatin) [7]. Finally, MVA is phosphorylated twice and decarboxylated to yield IPP/DMAPP. Alternatively, the MEP pathway is present in plant plastids, bacteria, and some protozoa groups, including apicomplexan parasites [2,6]. The MEP pathway starts with the condensation of glyceraldehyde 3-phosphate with pyruvate to form 1-deoxy-D-xylulose-5-phosphate (DOXP). This step is catalyzed by the 1-deoxy-D-xylulose 5-phosphate synthase (DXS). DOXP is next transformed to MEP by the enzyme 1-deoxy-D-xylulose 5-phosphate reductase (DXR). Importantly, DXR can be inhibited by fosmidomycin, which could thus be considered to be the homolog of statins on the MEP pathway. Remarkably, fosmidomycin has been proposed as an antibiotic for several protozoa and bacterial diseases [8,9]. Next in the pathway, MEP suffers modifications, including its condensation with CTP, a subsequent condensation with an ATP molecule, and a reduction. After these steps, hydroxymethyl-butenyl pyrophosphate (HMBPP) is produced. Later, this metabolite is converted into IPP/DMAPP by the enzyme hydroxymethyl-butenyl diphosphate reductase (LytB).

Regardless of the biosynthetic pathway, IPP and DMAPP can be interconverted by the IPP isomerases. In plants, IPP can be directly used for the production of phytohormones [10]. However, most isoprenoid-dependent pathways require isoprenic chains with lengths greater than one isoprenic unit. The first elongation step is the condensation of IPP and DMAPP to form geranyl pyrophosphate (GPP; 10 carbon), this step is catalyzed by the enzyme GPP synthase (GPPs). GPP can be condensed again with an IPP molecule to form farnesyl pyrophosphate (FPP; 15 carbon), with the help of the FPP synthase (FPPs), the enzyme target of bisphosphonates (e.g., alendronate, risedronate), which is widely used as a drug in the treatment of osteopenia and osteoporosis. Similarly, the geranylgeranyl pyrophosphate synthase (GGPPs) catalyzes the production of geranylgeranyl pyrophosphate (GGPP; 20 carbon) by condensing FPP with one more IPP molecule. After the formation of GGPP, several enzymes continue to extend the isoprenic chain to form longer structures: for example, octaprenyl pyrophosphate (8 isoprenic units); nonaprenyl pyrophosphate (9 isoprenic units; also known as solanesyl pyrophosphate); polyprenols of 13–21 isoprenic units; or even longer ones, such as natural rubber (>300 isoprenic units) [11,12]. Isoprenic moieties can suffer various chemical modifications. For example, in most photosynthetic organisms, hydrogenation of GGPP to phytyl pyrophosphate (phytyl-PP) is known to occur. Later, this phytyl-PP serves as a precursor for chlorophyll, tocopherols (vitamin E), and phylloquinone biosynthesis (vitamin K1) [13]. Another example is the reduction polyprenols, of 13–21 isoprenic units, to dolichols in eukaryotic organisms. Dolichols are phosphorylated to dolichyl-P by a dolichol kinase and then mainly employed as lipid anchors for sugar transport in the eukaryotic protein glycosylation and GPI biosynthesis pathways [14,15].

In any case, all polyprenyl pyrophosphates cited previously are employed as substrates for diverse anabolic pathways. For example, FPP and GGPP can be employed for an important post-translational modification of proteins. Typically, protein prenylation occurs when farnesyl or geranylgeranyl moieties are attached to soluble proteins, resulting in their anchoring to membranes. The most studied prenylated proteins are Ras, Rho, and Rap, which are small GTPases involved in cellular signaling and intracellular trafficking [16,17]. This post-translational modification is catalyzed by specific transferases that attach the FPP or GGPP moieties to the C-terminal cysteine residues of proteins containing the conserved motif for prenylation CAAX (C = cysteine, A = aliphatic amino acid, X = diverse terminal residue) [16,17]. Interestingly, alternative isoprenoid moieties were identified when linked to proteins. For example, in plants, a phytylation of proteins also occurs, in this case using phytyl-PP instead of FPP or GGPP [18], and protein dolichylation occurs in malaria parasites and human colon carcinoma cells [19,20]. Besides protein prenylation, polyprenyl pyrophosphates also play a role as constituents of membranes in plants [21,22,23], are required for the formation of menaquinone in bacteria (vitamin K2; requires GGPP) [24], and in the biosynthesis of sterols, like cholesterol in animals (requires FPP) [25]. Finally, polyprenyl-PPs also plays an important part in respiratory processes and lipoperoxidation defense. These molecules are necessary for the biosynthesis of ubiquinones (coenzyme Q), a metabolite that forms part of the electron transport chain at the mitochondrial membrane and an antioxidant cofactor elsewhere. The biosynthesis of ubiquinones involves the condensing of a benzoquinone ring with a polyprenyl-PP moiety of 6–10 isoprenic units; the actual length of the isoprene depends on the organism [26,27].

Isoprenoid biosynthesis is the target of some of the most prescribed drugs worldwide, as already cited. For example, statins are employed to treat hypercholesterolemia, as they inhibit mammalian biosynthesis of cholesterol, while bisphosphonates are used in the treatment of osteoporosis due to their ability to accumulate in bone mineral, inhibiting osteoclast activity [7,28,29]. Finally, fosmidomycin has been purposed as an antibiotic for the treatment of MEP-pathway-dependent protozoa and bacteria [8,30]. Moreover, isoprenoid biosynthesis inhibitors are under study for the treatment of many other diseases, which will be detailed below. An overview of the different pathways in isoprenoid metabolism and drugs discussed in this review is shown in Figure 1.

As explained above, the natural substrates for the biosynthesis of several cellular metabolites are the polyprenyl-PPs. However, their unphosphorylated counterparts, polyprenols (also referred to here as isoprenoid alcohols, despite their length), are also widespread, from bacteria to human cells [31,32,33,34]. Plants are probably the group of organisms where these metabolites have been most extensively studied [34]. Plants such as the acacia tree, *Acacia caven*, contain great amounts of farnesol (FOH) [35]; geranylgeraniol (GGOH) is particularly abundant in annatto (*Bixa orellana*) [36], as is solanesol in *Nicotiana tabacum* [37], and polyprenols of 13 isoprenic units in conifer extracts [38]. Medium-length isoprenoid alcohols of 3–4 isoprenic units in length serve as components of plant membranes [23,39]. However, just a few biosynthetic pathways seem to employ directly unphosphorylated polyprenols. For example, FOH is used to synthesize the hormone farnesoic acid by *Diploptera punctata* (Pacific beetle cockroach) [40], and, in most mammals, dietary phytol is degraded to phytanic acid for its excretion [41]. Similarly, both FOH and GGOH can be catabolized by mammals to farnesal and geranylgeranial, and subsequently to farnesoic acid and geranylgeranoic acid and some prenyl dicarboxylic acids. The enzymes for this pathway had already been identified as an excretion mechanism of polyprenols [32,42]. However, the most studied function of both FOH and GGOH in mammals is their role as metabolic regulators. Thus, FOH and GGOH possess a great ability to promote the degradation of the enzyme HMG-CoA reductase, resulting in a reduction in the MVA pathway, with concomitant reductions in the production of cholesterol or steroidogenic hormones [43,44].

## 2. The Origin of Geranylgeraniol and Farnesol

FOH and GGOH are also produced from GGPP and FPP by cells through the action of (already identified) specific phosphatases in mammals [45,46,47,48], plants [49], and insect tissues [50]. In mammalian-cultured cells, it has been observed that the overexpression of these phosphatases substantially decreases the prenylation of proteins and, as a result, induce cell growth defects, dysregulate the MVA pathway via degradation of the enzyme HMG CoA reductase, affect the organization of the cytoskeleton, and deregulate Rho proteins [45,46]. Therefore, it has been suggested that polyprenyl-PP phosphatases contribute to a feedback mechanism involved in maintaining optimal isoprenoid intracellular levels. In addition to endogenous FOH and GGOH, their dietary sourcing in mammals has been recently demonstrated [51,52]. Several foods, especially those of plant origin (e.g., sunflower oil, coconut oil, olive oil, tomato, or lettuce), display a high level of FOH and GGOH, which could be incorporated into animal cells after ingestion [52]. In addition to these, GGPP and FPP are constituents of all living organisms; thus, they are expected to be present in most foods. However, the pyrophosphate moiety of these compounds is unstable under acidic conditions and, therefore, are likely to be absorbed in digestion as isoprenoid alcohols [53]. Despite their importance, the reality of their dietary uptake, and their modulatory effects on biosynthesis, there is still no precise quantification of FOH and GGOH in human tissues (based on data from the Human Metabolome Database site, http://www.hmdb.ca/. Last accessed on 11 August 2022). For these reasons, and others that will be detailed in the next pages, they will be considered as nonessential nutrients of rising biomedical interest.

## 3. Geranylgeraniol and Farnesol Are Incorporated into Cellular Components

Both FOH and GGOH show several bioactive and clinical properties, among them, antibacterial [54], anti-tumorigenic [55], anti-inflammatory [56], and neuroprotective action [57,58], in addition to the regulation of hormone production [59]. However, the greatest biological activity of FOH and GGOH is related to their pyrophosphate derivatives, not to the alcoholic forms, i.e., their ability to provide a bypass to the isoprenoid biosynthetic pathway. In most cellular models, FOH and GGOH have a great ability to protect the cells from pharmacological or genetic-induced defects in isoprenoid metabolism. However, not all defects have been observed to be rescued. For example, there is no evidence that defects in cholesterol biosynthesis can be rescued by supplementation with GGOH. As was to be expected, it seems that the FOH- and GGOH-rescue effects are restricted to defects in enzymes of isoprenoid metabolism that are prior to the formation of the respective pyrophosphate forms of FPP and GGPP in biosynthesis. Considering this, GGOH has been showed to partially restore statin-induced contraction fatigue in mammalian muscle cells [60], prevent cytotoxicity in bisphosphonate-treated oral mucosa cells and fibroblasts [61,62], or ameliorate the phenotypes associated with a deficiency in MVA kinase in murine models and human cultured cells [63,64]. Furthermore, GGOH is required to maintain endotoxin tolerance in macrophages [65] and reverses the toxic effects of mevastatin in monocytic human THP-1 cells, without affecting its ability to reduce cholesterol synthesis [66]. Stemming from this, several authors have employed radio-labelled polyprenols to assess their possible incorporation into proteins. For example, the incubation of apicomplexan parasites, mammalian, or plant cells with either radio-labelled FOH or GGOH, has presented the incorporation of radioactivity into proteins [67,68,69,70]. Besides protein prenylation, evidences exist for the incorporation of radio-labelled FOH or GGOH into ubiquinone [71,72,73], dolichols [74], and steroids such as cholesterol [71,72,75], in different organisms. Interestingly, it has been observed that protein farnesylation and/or geranylgeranylation are the major isoprenoid-dependent processes for short-term viability in some organisms. Specifically, the biological importance of protein prenylation has been studied through metabolomics, drug-rescue assays and microscopic techniques in yeasts [76], in malaria parasites treated with isoprenoid biosynthesis inhibitors [77], as well as in the Chinese hamster ovary cell line UT-2, that has a defect in HGM-CoA reductase and is auxotrophic for MVA [78]. In all these cases, cell viability was not readily affected by deficiencies in ubiquinone, cholesterol, or dolichol biosynthesis, but solely by the protein prenylation defects.

## 4. The Geranylgeraniol and Farnesol Salvage Pathway

Despite the FOH and GGOH metabolization by several organisms, the natural substrates of all characterized polyprenyl transferases and synthases are polyprenyl pyrophosphates. Further, no evidence exists for this kind of enzyme being able to use polyprenols or any other polyprenyl derivative. This is likely due to the different orientation of these two types of lipids in the membrane. While polyprenols could be oriented parallel to the plane of the membrane, polyprenyl pyrophosphates strongly favor a perpendicular orientation, with their pyrophosphate moieties fiscally available for interaction with polyprenyl transferases and synthases [1,79]. Therefore, the incorporation of FOH and GGOH into the major isoprenoid pathways has been proposed to occur by the existence of a polyprenol salvage pathway acting via phosphorylation [76]. To the best of our knowledge, the first evidence for the existence of this pathway was presented in 1994 by Inoue et al. The authors observed CTP-mediated monophosphorylation of radio-labelled FOH in the 100,000 g membrane pellet from the microalga *Botryococcus braunii* [80]. The same activity was also observed in membrane extracts of the archeobacterium *Sulfolobolus acidocaldarius* two years later [81]. In this case, monophosphorylalation of radio-labelled GGOH was detected using a mixture of CTP, ATP, GTP, and UTP. Furthermore, a second phosphorylation, producing GGPP from GGP, was also detected using cytosolic fractions from *S. acidocaldarius*. The first evidence of this pathway in mammals was reported in 1998 by Bentinger et al. [82]. These authors observed farnesyl monophosphate (FP) being enzymatically synthesized from FOH in the 10,000 g supernatant of rat liver homogenates. Although this activity was observed with several nucleotides, the greatest activity was observed using ATP. This FOH kinase activity was located in rough and smooth microsomes, and associated with the inner, luminal surface of the vesicles. Further analyses in this system identified an activity able to phosphorylate FP to FPP. In this case, the catalytic activity was specific to CTP and probably localized on the cytoplasmic outer surface of microsomal vesicles. Alterations in FOH kinase and FPP phosphatase activities were observed in whole liver homogenates from rats receiving cholesterol- or cholestyramine-rich diets, compounds known to downregulate or up regulate the MVA pathway, respectively [83]. Remarkably, only FOH kinase was decreased in whole liver homogenates from rats on a cholesterol-rich diet. This suggests that FOH salvage can be inhibited if cholesterol requirements are satisfied.

In plants, phosphorylation of FOH and GGOH to their respective mono- and diphosphates was first observed in *Nicotiana tabacum* cell cultures [70]. In this case, the authors indicated a mechanism involving two successive monophosphorylation reactions, since radio-labelled CTP was formed when microsomes were put in contact with [^3^H] CDP and FPP or GGPP, but not with FP. Similar to this, a phytol salvage pathway was described in *Arabidopsis thaliana* [84]. The authors assessed if phytol could be phosphorylated to phytyl phosphate (phytyl-P) and phytyl-PP by two kinases located at the envelope membranes of plastids. The first phosphorylation was found to be CTP dependent, while the second one could be mediated by either CTP, GTP, UTP, or ATP. Our group also observed similar polyprenol phosphorylation reactions in malaria parasite extracts (*Plasmodium falciparum*); in this system, most probably, those reactions could serve as a mechanism to recycle polyprenols for the biosynthesis of parasitic cofactors. Furthermore, we observed parasites that could use these phosphorylated isoprenoid alcohols, including in Ras and Rap protein prenylation, in addition to condensing them into longer polyprenols and dolichols [85].

## 5. The Few Polyprenol/Polyprenyl-P Kinases Discovered

The consensus, thus far, is that the polyprenol salvage pathway is composed by two independent enzymes: a polyprenol kinase and a polyprenyl phosphate kinase [72]. However, to date, only a handful of genes have been unequivocally identified to encode this kind of enzyme; in *Solanum lycopersicum* plants, the *FOLK* and *VTE5* genes encode FOH kinase and phytol kinase, respectively [86]. The *Synechocystis* sp. gene Sc*VTE5* encodes a phytol kinase [87]. Finally, in *A. thaliana*, the At*FOLK* gene encodes a FOH kinase [88], namely, the At*VTE5* gene, by encoding a plastidic phytol kinase [87,89], and the recently described gene At*VTE6*, by encoding a plastidic phytyl-P kinase [90]. In all these cases, the enzymes require divalent cations and, although they can use different nucleotide trisphosphates, they showed a strong preference for CTP [87,88,89,90]. Concomitantly, some evidences on the great biological importance of the polyprenol salvage pathway in plants emerged. First, At*FOLK* gene expression was shown to be negatively regulated by the phytohormone abscisic acid. As a consequence, loss-of-function-*FOLK* mutants displayed an abscisic acid-hypersensitive phenotype and anomalous flower growth under abiotic stress, with the development of supernumerary carpels [88]. On the other hand, defects in the *VTE5* and *VTE6* genes in plants produced accumulations of free phytol and deficiencies in phytyl-PP-dependent pathways, such as tocopherol and vitamin K1 biosynthesis, as well as photosystem I instability and the reduced tolerance to abiotic stress [87,89,90]. All of these results suggest that de novo phytyl-PP biosynthesis may not be the major source of phytol for vitamin E and K1 biosynthesis in plants, but the remobilization of phytol from chlorophyl degradation.

In terms of structure, these genes and their encoded proteins show similarities and differences which must be commented on. The first of the genes reported, *VTE5*, was identified in mutagenesis studies of plants defective for vitamin E and K1 biosynthesis. *FOLK* was identified to encode a FOH kinase by similarity with *VTE5* [88]. On the other hand, both *VTE5* and *FOLK* gene products seem to be related enzymes, despite their different functions. Moreover, both show a modest similarity to dolichol kinase [90]. Remarkably, the putative function of *VTE6* as a phytyl-P kinase was first suggested when an analysis of similarities with bacterial genomes was carried out [91]. This gene showed similarities to those genes after putatively encoding for missing steps in isoprenoid metabolism, particularly with COG1836 sequences from photosynthetic prokaryotes. These genomic comparison tools could help in future identifications of polyprenol kinase genes in other organisms.

The biochemical characterization of these proteins was not straightforward. The identified polyprenol kinases needed to be recombinantly expressed in *S. cerevisiae* and *E. coli*, organisms where polyprenol and polyprenyl-P kinase activities are not naturally found [87,88,90]. Assays were performed in membrane extracts and required the use of radioactive-labelled substrates, followed by chromatography. Exceptionally, the recombinant expression of phytol kinase and phytyl-P kinase in *E. coli* allowed for phytol-feeding experiments, followed by the analysis of the product by mass spectrometry [87,90]. Advances in techniques using stable isotope labelling would be most beneficial in this field. Unfortunately, to date, no such assays have been reported for these kind of enzymatic activities.

## 6. Biomedical Importance of the Geranylgeraniol and Farnesol Salvage Pathway

As mentioned above, the polyprenol salvage pathway was first observed more than 25 years ago, but the enzymes responsible in animals or any pathogen are still unidentified [72,80]. Despite this, the interest in this pathway for the treatment of several human pathologies, ranging from osteonecrosis to infectious diseases, is increasing [92,93]. As a consequence, it is impractical to mention here everything published in this field, and we needed to make a selection of those reports that best illustrated the topics discussed, with a focus on biomedical problems.

### 6.1. Oncology

Oncology is probably the main area where polyprenols have been found to exert a dramatic role. This is somewhat unsurprising since isoprenoids are required for several cancer-related processes such as the Hedgehog pathway, the nuclear localization of the Yes-associated protein, steroid hormone formation, and transcriptional co-activation with the PDZ-binding motif. Furthermore, protein prenylation appears to be strongly related to oncogenesis and post-translational modification of growth-related oncoproteins such as Ras, Rac, and the Rho GTPase family [94]. Among all these proteins, alterations in Ras genes (including the oncogene *KRAS*) are very common in cancer, where they are present in approximately 22% of cases of these pathologies [95,96]. The search for treatments that effectively regulate this type of altered proteins is still on-going and thus, the issue is known as “the Ras problem”, and represents the focus of the NIH Ras Initiative (https://www.cancer.gov/news-events/cancer-currents-blog/2015/turning-off-broken-switch, last accessed on 7 July 2022). Among the strategies devised, inhibitors of the MVA pathway for the treatment and prevention of different types of cancers have been explored for decades. However, the use of statins mostly shows an unsatisfying ability to treat and prevent cancer [51,97,98,99]. Remarkably, it has been shown that GGOH can limit the antineoplastic activity of statins against some types of cancer, such as ovarian carcinoma and neuroblastoma [51,98]. Furthermore, de Wolf et al. observed extracts from foods rich in GGOH, such as vegetable oils, being able to block pitavastatin-induced apoptosis in ovarian cancer cells, suggesting that diet may influence the outcome of clinical trials. In agreement with this, oral administration of pitavastatin has caused regression of ovarian tumour xenografts in mice maintained on a GGOH-free diet. However, when their diet was supplemented with GGOH, pitavastatin failed in reducing tumor growth. This is a strong indication that GGOH may be naturally, exogenously incorporated and bioactive in animals. Recently in 2022, GGOH-deficient diets have been designed to make statins viable for the treatment of some types of human cancers [52]. Interestingly, polyprenols alone have long been known to affect tumor cell proliferation [55,100], but the interest in these compounds has not dwindled. Recently, the application of low doses of farnesol was observed to reduce proliferation of lung cancer A549 and colon adenocarcinoma Caco-2 cell lines, compared with a healthy human lung epithelial BEAS-2B cell line, but, surprisingly, at high concentrations the situation reversed [101]. The antioxidant property of farnesol may be one of the reasons for this antitumoral activity, as it has been shown to protect rats against diethyl nitrosamine-induced hepatocellular carcinoma [102].

### 6.2. Infectious Diseases

The use of MVA pathway inhibitors as antibiotics and antiviral agents was also explored in several reports. Among the organisms targeted, there were viruses (e.g., respiratory syncytial virus, murine cytomegalovirus, gamma herpesvirus and influenza virus), protozoa (e.g., *Leishmania donovani*, *Leishmania major* and *Toxoplasma gondii*), fungi (e.g., *Candida* spp. and *Cryptococcus neoformans*), and bacteria (e.g., *Listeria monocytogenes*, *Coxiella burnetiid*, *Rickettsia conorii* and *Staphylococcus aureus*) [103,104,105]. In general, statins compromised the pathogen’s ability of immune evasion and survival, associated with a deficit of host-derived MVA pathway intermediates [103,104,105]. Similarly, the MEP-pathway has also been found to be an interesting drug target against *Plasmodium falciparum* [30,106,107,108], *Toxoplasma gondii* [109], *Babesia microti* [110], or *Mycobacterium tuberculosis* [111]. Furthermore, fosmidomycin has already been evaluated in several clinical trials as a promising treatment for malaria, albeit with limited success [112]. Importantly, most MEP-dependent pathogens can also be rescued from fosmidomycin by FOH and GGOH [77,109,110,111,112,113,114]. Therefore, the incorporation of exogenous polyprenols seems to be a possible mechanism limiting the efficacy of treatments against many pathogens and cancer. It could be envisaged, then, that pharmacological inhibition of FOH/GGOH phosphorylation could be of interest to potentiate the effects of isoprenoid biosynthesis inhibitors in a therapeutical setting.

### 6.3. Isoprenoid Deficiencies

The polyprenol salvage pathway plays a role in several conditions etiologically related to isoprenoid deficiencies. These deficiencies may occur as a side effect of pharmacological treatment, during aging, due to a genetic disorder or idiopathic reasons. For exemple, osteoporosis is a bone calcification deficiency that correlates with aging and is usually treated with biphosphponates [115]. Osteonecrosis of the jaw is the most severe adverse effect in patients on bisphosphonates [116,117]. At therapeutical doses, bisphosphonates decrease the levels of prenyl pyrophosphates available for the prenylation of GTPases in osteoclasts, inhibiting bone resorption [28,29]. However, if in excess, bisphosphonates can also uncouple bone remodeling and healing, leading to osteonecrosis. In this context, isoprenoids, with GGOH among them, have been shown in in vitro and in vivo models to be able to restore GTPase prenylation and, thus, to be potentially useful in preventing and treating osteonecrosis of the jaw [116,117].

Statins, used to manage dyslipidemias, also possess several side effects, including headache, dizziness, discomfort or weakness in skeletal muscles, constipation, diarrhea, and hepatotoxicity [118]. Specifically, hepatotoxicity and muscle pain are the most severe side effects, and are associated with deficiencies in protein prenylation [94,119,120]. Furthermore, long term treatment with statins has been shown to decrease dolichol and ubiquinone levels and, as a consequence, makes organisms more susceptible to oxidative damage, and to be inefficient in protein post-translational modifications and in obtaining energy via oxidative phosphorylation [94,119,120]. Despite statin side effects being rarely severe, at times it is necessary to stop the therapy, exposing the patient to an increased cardiac risk [118]. With this in mind, the administration of ubiquinone complements was first proposed as a strategy to ameliorate statin side effects [121,122]. However, these complements were not demonstrated to satisfactorily reduce those major side effects, and ubiquinone was poorly orally absorbed [121,122]. Recently, commercial GGOH rich extracts and preparations were launched that have shown promising effects of restoring protein prenylation in this context [94,119,120]. In particular, GGOH supplements supported a beneficial effect against statin-induced myopathy and in the prevention of skeletal muscle fatigue in preliminary studies [119,120]. It must be noted that polyprenols could also be effective against statin-induced insulin resistance in muscle tissue [123] and against the dysregulation of hepatic glucose metabolism [124]. Furthermore, polyprenol-rich extracts of >13 isoprenic units was also commercialized. Once incorporated, these polyprenols were probably converted into dolichols, and then phosphorylated to dolichyl-P by dolichol kinase. In this case, however, dolichols were seen to have no protective effect against pitavastatin [49], which is in agreement with them being uninvolved in protein prenylation. Conceivably, another possible strategy to increase physiological levels of FPP and GGPP in statin-treated patients could be the inhibition of polyprenyl-PP phosphatase. However, this possibility remains unexplored experimentally.

### 6.4. Dyslipidemias

It could be envisaged that FOH may promote dyslipidemias, as some studies demonstrated that this compound is converted into cholesterol in human cells due to the polyprenols salvage pathway (a metabolic route presumably not regulable with statins [72,75]). However, the contribution of FOH to high cardiovascular risk has not been proven, while the opposite effect has been reported. Abukhalil et al., showed that administration of FOH attenuated oxidative stress and liver injury, at the same time that it modulated fatty acid synthase and acetyl-CoA carboxylase in high cholesterol-fed rats [125]. Similarly, long chain polyprenols and phytol have been shown to have a direct effect as lipid-lowering compounds [126]. Mechanistically, FOH would act similarly as a statin, downregulating MVA pathway, at the same time as the fatty acid synthase and acetyl-CoA carboxylase, while preserving protein farnesylation. Therefore, FOH itself may be an interesting novel strategy for the treatment of dyslipidemias.

### 6.5. Rare Diseases

Mutations in the human mevalonate kinase gene are the cause of a short spectrum of related rare diseases. Mevalonate kinase deficiency (ORPHA code 309025; see www.orpha.net, last access: 21 October 2022) is caused by hypomorphic mutations, and the patients typically display fever episodes and hyperimmunoglobulinemia [127], while complete loss of function of the *MVK* gene brings about mevalonic acidura (ORPHA code 29; see www.orpha.net, last access: 21 October 2022), a severe form of the condition associated with short life expectancy and dysmorphic features. Since the major effects of these diseases are also related to defects from isoprenoids biosynthesis and protein prenylation, the dietary supplementation with polyprenols has been tested as a treatment for mevalonate kinase deficiency. Again, the administration of geraniol, FOH, and GGOH improved symptoms and inflammation markers both in *in vitro* and *in vivo* models [63,128,129]. Interestingly, polyprenols could also act as anti-inflammatory agents per se [56], as well as being effective compounds to promote neuroprotection [57,58] or regulate hormone production [59]. What is especially interesting is the GGOH ability to attenuate lipopolysaccharide-induced inflammation in rats and mouse-derived microglial cells, a phenomenon strongly associated with several neurodegenerative diseases [130,131]. While some of these effects of polyprenols may one day be ascribed to its salvage, many may be related to other characteristics, such as their intrinsic ability to reduce oxidative stress [132]. Thus, this falls outside the scope of this review.

## 7. Conclusions and Perspectives

In animals, both FOH and GGOH can be enzymatically synthetized or incorporated from the diet, producing diverse biological effects. Most effects are the consequence of the existence of a pathway for polyprenol salvage, composed by a polyprenol kinase and a polyprenyl-phosphate kinase. Once FOH and GGOH are pyrophosphorylated, they are incorporated into the protein prenylation process, as well as in other isoprenoid-dependent anabolic pathways, such as the synthesis of ubiquinone, cholesterol, and dolichol. This FOH/GGOH salvage pathway seems to be active in bacteria, animals, and plants as a mechanism to regulate or by-pass the isoprenoid biosynthetic pathway. However, to date, just a few enzymes have been identified, only in plants and cyanobacteria. Most likely, the study of this pathway has suffered from technical problems: first, these enzymes seem to present little sequence similarity to others with like functions, and no easy complementation assays are available. Furthermore, polyprenyl kinases are presumably membrane proteins and thus, are difficult to express heterologously. Enzyme activity and cellular polyprenol-distribution assays are cumbersome, since they usually require radio-labelling techniques, which are nowadays much in disuse. As a consequence, the enzymes responsible for this pathway in animals or any pathogen are still unidentified. However, considering the evidence, it seems patent that the identification of these enzymes could shed light on several infectious diseases, cancer, dyslipidemias, nutritional deficiencies, and the secondary effects of several drugs. Throughout this text, we aimed to present, in a simplified but comprehensive way, the main data on the FOH and GGOH salvage pathway, emphasizing its biomedical importance. Therefore, the main goal of this review was to raise awareness among the scientific community of the need for a better study of the polyprenol salvage pathway, and to discover the corresponding kinases in animals and pathogens.

## Figures and Tables

**Figure 1 molecules-27-08691-f001:**
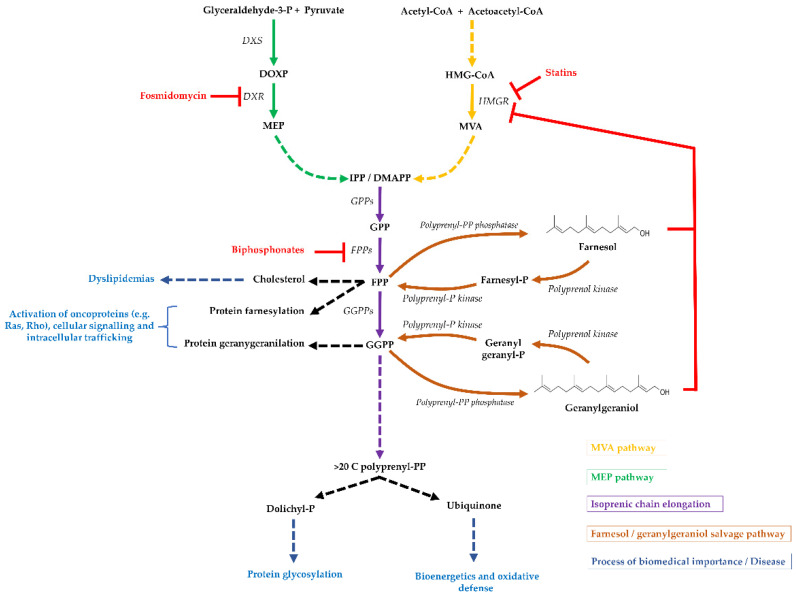
**Isoprenoid biosynthesis in cells.** A simplified view of isoprenoid biosynthesis pathways and the targets of different drugs. Although shown as a general situation, some parts may not be present in all organisms. For example, the MVA pathway (light blue lines) is present in mammals and the cytosol of plants; the MEP pathway (green lines) is found in the plastid of plants and apicomplexan parasites, but not in mammals; the geranylgeraniol and farnesol salvage pathway (brown lines) is not present in *Saccharomyces cerevisiae* or *Escherichia coli*. Continuous arrows indicate a single enzymatic step, discontinuous arrows indicate multiple steps and blue arrows/text refers to processes of biomedical importance or diseases. Red arrows indicate a drug inhibition or metabolic downregulation. The chemical structures of FOH and GGOH are also shown. Abbreviations are as in the text.

## Data Availability

Not applicable.

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
