# Peer review of "The Biomedical Importance of the Missing Pathway for Farnesol and Geranylgeraniol Salvage"

_molecules, 2022, doi:10.3390/molecules27248691_

Round 1

Reviewer 1 Report

Regarding article titled The clinical importance of the missing pathway for farnesol and geranylgeraniol salvage. Following are some of my personal opinions:

1. It is suggested that the author's description should be more concise and clear in the abstract

2. It is suggested to supplement the introduction of the article

3. The title of the article is the clinical importance of the missing pathway for farnesol and geranylgeraniol salvage, but the clinical significance is rarely described in the article, and minimize the plant aspects. it is recommended to focus on the clinical significance of the supplement.

4. The idea of the article is not clear enough. it is recommended that the author sort out the overall structure of the article and clarify it

5. It is recommended to supplement the content of the mechanism diagram so that readers can understand it more fully.

Author Response

Please, find responses to reviewers in the PDF file attached

Reviewer 2 Report

MANUSCRIPT: 2058029

TITLE: The clinical importance of the missing pathway for farnesol and geranylgeraniol salvage

The manuscript 2058029 “The clinical importance of the missing pathway for farnesol and salvage geranylgeraniol”, presents a review of the literature.

The manuscript presented is well structured.

The review is clearly written, well systematized and comprehensive for the topic, and literature cited is adequate and most of the papers cited are from the last five years.

Similar reviews are not known, and it is of interest to the scientific community.

The conclusions are consistent and in accordance with the quotes listed.

I just have a small request to the authors:

References must be presented in accordance with the Reference List and Citations Style Guide for MDPI Journals. Please download the full MDPI Reference List and Citations Style Guide at MDPI | Reference List and Citations Style Guide and proceed as per the document and present the references as per the document and include the doi and/or ISBN.

Author Response

(The authors gave the same response as above.)

Round 2

Reviewer 1 Report

Regarding article titled The clinical importance of the missing pathway for farnesol and geranylgeraniol salvage. Following are some of my personal opinions:

1、The Isoprenoid biosynthesis and distribution section suggests that the author describe it more concisely.

2. It is suggested to supplement the introduction of the article.

3. It is recommended to supplement the content of the mechanism diagram so that readers can understand it more fully.